# Fear Conditioning by Proxy: The Role of High Affinity Nicotinic Acetylcholine Receptors

**DOI:** 10.3390/ijms242015143

**Published:** 2023-10-13

**Authors:** Zinovia Stavroula Chalkea, Danai Papavranoussi-Daponte, Alexia Polissidis, Marinos Kampisioulis, Marina Pagaki-Skaliora, Eleni Konsolaki, Irini Skaliora

**Affiliations:** 1Center of Basic Research, Biomedical Research Foundation of the Academy of Athens, 11527 Athens, Greece; danaipapdap@gmail.com (D.P.-D.); mkampisioulis@gmail.com (M.K.); 2Master’s Program in Cognitive Science, National and Kapodistrian University of Athens, 15771 Athens, Greece; 3Athens International Master’s Program in Neurosciences, National and Kapodistrian University of Athens, 15772 Athens, Greece; 4American College of Greece Research Center (ACG-RC), 15342 Athens, Greece; apolissidis@bioacademy.gr; 5Center for Experimental, Clinical, and Translational Research, Biomedical Research Foundation of the Academy of Athens, 11527 Athens, Greece; 6University of York Medical School, Heslington, York YO10 5DD, UK; mkpagaki@gmail.com; 7Psychology Department, Deree-The American College of Greece, 15342 Athens, Greece; elenkons@gmail.com; 8Department of History and Philosophy of Science, National and Kapodistrian University of Athens, 15771 Athens, Greece

**Keywords:** β2 nicotinic receptors, cholinergic system, social interaction, flexible behaviour, individual variation, mouse models, prefrontal cortex, associative learning, classical conditioning, fear transmission

## Abstract

Observational fear-learning studies in genetically modified animals enable the investigation of the mechanisms underlying the social transmission of fear-related information. Here, we used a three-day protocol to examine fear conditioning by proxy (FCbP) in wild-type mice (C57BL/6J) and mice lacking the β2-subunit of the nicotinic acetylcholine receptor (nAChR). Male animals of both genotypes were exposed to a previously fear-conditioned (FC) cage mate during the presentation of the conditioned stimulus (CS, tone). On the following day, observer (FCbP) mice were tested for fear reactions to the tone: none of the β2-KO mice froze to the stimulus, while 30% of the wild-type mice expressed significant freezing. An investigation of the possible factors that predicted the fear response revealed that only wild-type mice that exhibited enhanced and more flexible social interaction with the FC cage mate during tone presentations (Day 2) expressed fear toward the CS (Day-3). Our results indicate that (i) FCbP is possible in mice; (ii) the social transmission of fear depends on the interaction pattern between animals during the FCbP session and (iii) β2-KO mice display a more rigid interaction pattern compared to wild-type mice and are unable to acquire such information. These data suggest that β2-nAChRs influence observational fear learning indirectly through their effect on social behaviour.

## 1. Introduction

Reacting promptly to the presence of threats is critical to survival, and the ability to identify cues that predict danger is essential in this process. Learning about potentially harmful events enables the formation of associations between external cues and emotional/motivational states such as fear. In turn, fear can be characterized by anxiety and agitation due to the expectation of impending danger and thus serves as an adaptive alert mechanism for the organism. It can be acquired either through direct experience or, indirectly, through social transmission [1]. (Olsson et al., 2007).

Much of our knowledge regarding the neurobiological mechanisms of fear learning comes from the extensive literature on Pavlovian (classical) fear conditioning. A typical fear conditioning paradigm involves the association of a neutral stimulus (conditioned stimulus, CS) with a naturally aversive stimulus (unconditioned stimulus, US), which elicits fear responses. After repeated temporal associations of the CS (e.g., tone) with the US (e.g., shock), the presentation of the tone by itself elicits a conditioned fear response (e.g., immobility). The consistency in the physiological expression of conditioned fear elicited by the basic protocol indicates that mechanisms of emotional learning are conserved across species [2] and neuropsychological studies in humans have replicated the results observed in animals [3]. 

In addition, humans routinely use a safer way to learn about threats, namely the social transmission of information about danger through language or the observation of fear signals in others. Recent studies have also documented the social detection and transmission of fear signals in other species, including birds, rodents, cats and non-human primates [3]. It has been suggested that fear signs alert the observer about potential danger and assign a threat value to the cue or the context associated with the threat [4]. In fear conditioning terms, a conspecific’s fear expression serves as the aversive stimulus (US), which evokes a fear response in the observer and becomes associated with a paired neutral stimulus (CS). Several indications suggest that observational fear learning relies considerably on the same basic associative learning processes as classical fear conditioning—although it seems to show greater interspecies variability [4]. Interestingly, the anterior cingulate cortex (ACC) is necessary for social fear learning but is not required for classical fear conditioning [5,6]. 

In this study, we focus on observational fear learning in rodents, since previous studies reveal that they are sensitive to the distress of others, and an encounter with a distressed conspecific can modulate how a rodent subsequently learns about environmental signals that predict fearful situations [7]. Research in mice has shown that these animals learn to fear a cue (or context) after observing a conspecific undergoing fear conditioning to that stimulus [6,8,9]. In addition, Jeon et al. (2010) demonstrated that the magnitude of the fear response depends on the relatedness or familiarity of the observer to the demonstrator, suggesting that social relationships are an important element in observational fear conditioning in mice [9]. Moreover, Chen et al. (2009) showed that the genetic background can influence the degree to which a mouse is responsive to the distress in others, since a fear response was recorded in C57BL/6J mice (B6) but not in BALB/cJ (BALB) [8]. Likewise, only B6 observer mice exhibited physiological correlates of empathy (namely, heart rate deceleration) while they were experiencing conspecific distress. The B6 and BALB mouse strains have been used by several laboratories as an experimental model of sociability, with B6 mice representing a sociable strain and BALB a possible mouse model of autism. Thus, B6 mice are an appropriate strain to explore the mechanisms underlying the possible association between observational fear learning and sociability.

In a different experimental paradigm, a series of studies in rats explored fear learning in the absence of a threat based only on the demonstrators’ fearful response to the CS. In this case, the demonstrator underwent fear conditioning to a tone and was subsequently presented with the tone in the presence of an observer [5,10,11,12,13]. This model (Fear Conditioning by Proxy) investigates observational fear learning when the demonstrator and observer are free to interact. Although Kim et al. (2010) showed that the observer rats acquired a freezing response to the tone only if they had prior fear experiences themselves (experience with unsignaled foot shocks) [10], the rest of the studies demonstrated that some observer rats with no direct fear experience expressed appreciable freezing responses when they were presented with the tone [10,11,12,13]. Interestingly, these latter studies demonstrated a positive correlation between the observers’ freezing response to the CS and the amount of social interaction with demonstrators during the observation session. Thus, a CS that predicted distress in a familiar demonstrator engendered a fear response only in the observers that had high levels of social interaction with the demonstrators.

Overall, the experiments considered above demonstrate that even rodents do not require direct experience of a CS–US presentation in order to express fear to a cue/context that predicts danger and implicates several variables that modulate rodent sensitivity to others’ distress, including genotype [8], familiarity [9,12] and the amount of social interaction [5,11,12,13]. In the present study, we aimed to assess the role of social interaction in fear conditioning by proxy in mice, as well as to explore the underlying neurobiological mechanisms. For this purpose, we decided to use genetically modified mice that exhibit impairments in social interaction and deficits in brain areas correlated with social interaction. Mice lacking the β2 subunit of nicotinic acetylcholine receptors (β2-KO nAChRs) presented as a good candidate.

nAChRs are ligand-gated ion channels widely distributed in the brain and implicated in modulating central nervous system functions [14]. Brain nAChRs are pentameric oligomers composed of protein subunits (12 subunits: α2–α10 and β2–β4) arranged in various combinations of α and β. The molecular characterization of these proteins and the use of transgenic animals have highlighted the role of nAChRs in cognitive functions [14,15]. Particularly, studies in mice lacking the gene encoding the β2 subunit (β2-KO) demonstrated an important role of these high-affinity receptors in cognitive and executive processes [16,17,18,19,20,21]. 

Mice lacking β2-containing receptors display morphological alterations in the neurons of the cingulate cortex (ACC and PrL) [22,23]—a brain area implicated both in social interaction (in humans, non-human primates and rodents) [24,25] and observational fear learning (in humans and mice) [1,5,6,26]. In addition, they exhibit increased social interaction after isolation, which is restored to normal levels after β2 re-expression in the prelimbic (PrL) area of the prefrontal cortex (PFC) [17]; and impaired behavioural flexibility, which resembles the effects of brain damage in the ACC and PFC [16,17,21,27,28]. Overall, β2-KO mice show impairments in the brain areas that have been correlated with social interaction and impaired behavioural flexibility and social interaction. For all these reasons, we decided to explore the role of social interaction in observational fear learning using these animals.

Specifically, we aimed to test three hypotheses using the fear conditioning by proxy (FCbP) paradigm: (1) Can male mice acquire fear associated with a CS based only on a conspecific’s fear response to that stimulus? (2) Does the social transfer of fear depend on the social interaction between observers and demonstrators? And (3) is social transfer of fear affected by the lack of the β2-nAChRs? We used B6 (wild-type and β2-KO) mice and formed each “observer-demonstrator pair” using animals of the same cage, in order to control for genotype and familiarity.

## 2. Results


**WT and β2-ΚO mice display similar fear conditioning during the FC session (day 1).**


The experimental protocol lasted three days for each mouse triad (see Section 4.2 and Section 4.4, as well as Figure 10 and Figure 11). During the FC session on the first day, one mouse of each triad (demonstrator, FC) underwent cued fear conditioning. Figure 1 presents the mean percentage of time that the animals of each genotype spent freezing during the five consecutive tone–shock pairings. A two-way mixed-design ANOVA was performed in order to examine if the β2-KO mice exhibited differences in cued fear acquisition compared to the WT group. “Tone-shock pairing” was the within-subject factor (5 levels: 1st, 2nd, 3rd, 4th and 5th tone), “genotype” was the between-subject factor (2 levels: β2-KO and WT) and “mean freezing” was the dependent variable. There was no significant interaction of genotype and tone–shock pairing (sphericity assumed; *F*(4, 104) = 0.788, *p* = 0.535, *η_p_*^2^ = 0.029) or a genotype main effect (*F*(1, 26) = 0.443, *p* = 0.512, *η_p_*^2^ = 0.017) on freezing duration. As expected, there was a significant main effect of tone–shock pairing on freezing duration (sphericity assumed; *F*(4, 104) = 0.443, *p* < 0.001, *η_p_*^2^ = 0.747). Multiple comparisons (Bonferroni adjustment) revealed that all the mice froze significantly more during the last three pairings compared to the first and second pairings (*p* < 0.001, for each comparison). Thus, **no differences were found between the β2-ΚO and WT mice in either the rate of cued fear learning or the magnitude of the fear response (duration of freezing), indicating that the β2-ΚO mice exhibited a normal acquisition of cued fear.**


**WT and β2-ΚO animals exhibit the same freezing behaviour during the FCbP session (day 2).**


Since the two genotypes displayed no difference in cued fear acquisition, it was expected that during the FCbP session on the second day, the β2-ΚO and WT demonstrators (FC animals) would display a similar fear-conditioned response in the presence of their cage mates (observers, FCbP). To examine if all the FC animals adopted the same freezing pattern during the entire FCbP session (i.e., both the pre-tone and tone phases), a two-way mixed ANOVA was performed. The “test-phase” was the within-subject factor (two levels: pre-tone and tone), and “genotype” was the between-subject factor (two levels: β2-KO and WT). A significant interaction between genotype and the test phase (*F*(1, 26) = 5.982, *p* = 0.022, η_p_^2^ = 0.187) was found. A further analysis showed no significant simple main effect of genotype either during the pre-tone (*F*(1, 26) = 1.761, *p* = 0.196, *η_p_*^2^ = 0.063) or tone phase (*F*(1, 26) = 1.302, *p* = 0.264, *η_p_*^2^ = 0.048). Instead, there was a significant simple main effect of phase on freezing time for both genotypes (WT: *F*(1, 13) = 143.249, *p* < 0.001, *η_p_*^2^ = 0.917 and KO: *F*(1, 13) = 53.962, *p* < 0.001, *η_p_*^2^ = 0.806), suggesting that all the FC mice froze significantly more during the tone phase (tone presentations plus ITIs). (Figure 2). These findings confirm that β2-KO mice show unimpaired fear conditioning and show that **the observers (FCbP) of both genotypes interacted with the demonstrators (FC) that displayed similar freezing patterns. Hence, we can be confident that the same fear information is available to the β2-KO and WT observers about the tone.**


**The WT mice displayed enhanced and more flexible social interaction compared to the KO animals during the FCbP session (day 2).**


Having established that the β2-ΚO and WT fear-conditioned animals have similar freezing behaviour, we next examined whether the two genotypes differed in the way they interacted during the FCbP session. First, we compared the *total* interaction time between the FC and FCbP animals in the two genotypes (Figure 3A) and found that the β2-KO observers spent significantly less time interacting with their FC cage mate (independent samples *t*-test; *t*(26) = 3.480, *p* = 0.002). Specifically, the β2-ΚO mice spent almost half the time in social interaction (WT: 12.79 ± 1.68% vs. β2-ΚO: 6.42 ± 0.73%).

To further explore this genotype difference and to examine whether the way the animals interacted was influenced by the CS, we quantified the duration of social interaction separately for each test phase (tone vs. pre-tone). A two-way mixed-design ANOVA was performed with “test-phase” as the within-subject factor (two levels: pre-tone and tone phase) and “genotype” as the between-subject factor (two levels: β2-ΚO and WT). The interaction between genotype and test phase was found to be statistically significant (*F*(1, 26) = 4.266, *p* = 0.049, *η_p_*^2^ = 0.141). An analysis of the simple main effects showed that the WT observers interacted significantly more than the β2-KO observers during the tone phase (*F*(1, 26) = 6.805, *p* = 0.015, *η_p_*^2^ = 0.207), but there was no such difference during the pre-tone phase (*F*(1, 26) = 2.501, *p* = 0.126, *η_p_*^2^ = 0.088). Importantly, only the WT mice increased their interaction levels during the tone phase (WT: *F*(1, 13) = 5.474, *p* = 0.036, *η_p_*^2^ = 0.296 vs. KO: *F*(1, 13) = 0.094, *p* = 0.763, *η_p_*^2^ = 0.007) (Figure 3B). **These results reveal that the β2-ΚO mice sustained low and stable levels of social interaction during the entire FCbP session, while the WT mice significantly enhanced their interaction levels during the tone phase.**


**The mean freezing behaviour during the cued fear test (day 3) is similar in FCbP and naïve animals of both genotypes.**


Having documented the freezing and social behaviour of all the animals during the first 2 days, we then asked the central question: can observer mice learn to fear a previously neutral stimulus based only on their interaction with a fear-conditioned cage mate? To do this, we examined (a) whether the FCbP animals expressed greater levels of freezing to the tone compared to the naïve controls and (b) whether there was a genotype effect on fear conditioning by proxy. This was assessed by a two-way independent-measures ANOVA. There were two between-subjects factors: “genotype” (two levels: β2-ΚO and WT) and “group” (two levels: FCbP and naïve). We found no significant interaction between genotype and group (*F*(1, 48) = 0.532, *p* = 0.469, *η_p_*^2^ = 0.011) nor a significant main effect of either genotype (*F*(1, 48) = 1.853, *p* = 0.180, *η_p_*^2^ = 0.037) or group (*F*(1, 48) = 0.048, *p* = 0.828, *η_p_*^2^ = 0.001) on tone freezing (Figure 4). In other words, **neither group of mice seemed to freeze to the tone**.


**WT-FCbP mice display two patterns of freezing during the cued fear test (day 3).**


Although there was no significant increase in the mean freezing to the tone for either genotype, we noticed that the WT-FCbP mice displayed a much greater variation in freezing compared to the WT-naïve animals (WT-FCbP SD = 9.948%, vs. WT-naïve SD = 2.967%) and that one third of the WT-FCbP group displayed considerable freezing, while the rest of the group did not (Figure 5A,B). Interestingly, previous studies had demonstrated a similar effect in rats (Bruchey et al., 2010; Jones and Monfils, 2016; Jones et al., 2018) and showed that higher freezing was positively correlated with a greater interaction during the FCbP session (day 2). In contrast, this “bimodality” in freezing behaviour was not evident in the β2-ΚO mice (KO-naïve SD = 5.16% versus KO-FCbP SD = 5.30%).


**The pattern of social interaction during the FCbP session can predict WT-FCbP freezing to the tone.**


The significant variability of the FCbP WT group was intriguing, as it suggested that some mice are able to acquire fear information from conspecifics while others are not. Hence, we decided to investigate the factors that underly the greater freezing variation within this group by exploring various aspects of the social interaction displayed by the WT animals (Table 1). In order to have a direct comparison to previous publications employing the FCbP protocol [5,11,12,13], we performed a stepwise multiple regression analysis on the WT FCbP freezing behaviour. The social interaction variables shown in Table 1 were assessed as potential predictors of the observers’ freezing response (criterion variable). Despite the modest sample size, all criteria for the analysis were satisfied with normality, linearity, homoscedasticity and tolerance > 0.1.

The statistical model revealed two variables as significant predictors of freezing behaviour (indicated by the stars in Table 1): (i) the percentage of time the observer spent in social interaction with the FC mouse throughout the duration of the 5 tone presentations (“interaction during tone”), and (ii) the *change* in interaction displayed between the two test phases (tone phase vs. pre-tone phase; “interaction change”). Both of them were positively correlated with WT-FCbP freezing (Figure 6), and together they accounted for 85.2% of the freezing variance in the group (*F*(2, 11) = 25.932, *p* < 0.001, *R*^2^ = 0.852). More specifically, 69.7% of the variance was explained by the change in interaction time (*R_change_* = 0.697) and an additional 12.7% by the “interaction during tone” (*R_change_* = 0.127). Notably, **the “interaction change” alone was the most important predictor, indicating that flexibility in social interaction is the essential factor**.

To visualize the link between social interaction and freezing behaviour for the WT FCbP animals, we plotted the two predictor variables against each other (Figure 7A). The horizontal and vertical black lines represent the mean interaction during the tone (12.33) and mean interaction change (5.75), respectively. Mice that displayed high levels of interaction during the tone presentations *and* increased interaction after the first tone are clustered in the upper right quadrant (blue triangles). These define the WT FCbP+ subgroup. All the other mice that do not fulfil these two criteria are scattered in the other three quadrants (blue circles) and comprise the WT FCbP− subgroup. As illustrated in Figure 7A, in contrast to the “tone dependent interaction pattern” manifested by the WT FCbP+ subgroup, the WT FCbP− mice displayed quite stable interactions throughout the FCbP session and/or low interactions during tone presentations. **Based on the results of our regression model, only the WT observers in the FCbP+ group would be expected to acquire observational fear learning and display freezing to the tone on day 3 (cued fear test).**

To test this prediction, the mean freezing time between the three WT subgroups (naïve, FCbP− and FCbP+) was analysed using a two-way mixed-design ANOVA. “Test-phase” was the within-subject factor (two levels: pre-tone and tone), and “group” was the between-subject factor (3 levels: FCbP+, FCbP− and naïve) (Figure 8). A significant interaction between the test phase and group was detected (*F*(2, 21) = 7.301, *p* = 0.004, *η_p_*^2^ = 0.410), indicating that the WT mice belonging to the different subgroups had different freezing responses. There were also significant simple main effects of the group on pre-tone freezing (*F*(2, 21) = 4.196, *p* = 0.029, *η_p_*^2^ = 0.286), with the FCbP+ animals freezing 9.625 ± 3.340% more than the FCbP− animals, and on tone freezing (*F*(2, 21) = 39.283, *p* <0.001, *η_p_*^2^ = 0.789), where the FCbP+ mice froze significantly more than the FCbP− or naïve animals. Multiple comparisons (Bonferroni adjustment) demonstrated that the FCbP+ group was the only one that showed significantly higher freezing during the tones compared to the pre-tone phase (*F*(1, 21) = 18.361, *p* <0.001, *η_p_*^2^ = 0.466). **These findings indicate that the WT-FCbP+ mice expressed conditioned fear to the CS, while the animals in the WT-FCbP− and naïve groups did not. Thus, only a tone-dependent interaction pattern seems to enable fear conditioning by proxy.**


**None of the KO-FCbP mice displayed the tone-dependent interaction pattern needed for the social transmission of cued fear.**


The two significant social interaction variables that reflect the animals’ interaction pattern and influence freezing on day 3 were further compared between the two WT FCbP subgroups and β2-KO FCbP mice. As shown in Figure 9, the WT-FCbP+ mice had significantly enhanced duration of interaction with the FC mouse during the tone phase (Figure 9A) and also exhibited higher levels of interaction during the tone presentations (Figure 9B) compared to both the WT-FCbP− and β2-ΚO FCbP groups (*F*(2, 21) = 16.630, *p* < 0.001, *η_p_*^2^ = 0.613 with mean interaction change = 13.00 ± 2.73% and *F*(2, 21) = 6.573, *p* = 0.006, *η_p_*^2^ = 0.917 with mean interaction during tone = 20.75 ± 3.35%). In contrast, the FCbP− mice displayed a tone-independent interaction pattern, with quite stable and/or low interactions during the tone (mean interaction change = 2.13 ± 0.95%, and mean interaction during tone = 8.13 ± 2.68%). Similarly, the β2-KO observers displayed tone-independent interaction, with low levels of interaction during the tone phase (mean interaction change = −2.17 ± 1.50%, and mean interaction during tone = 6.42 ± 1.90%).

Altogether, our results indicate that for WT animals, fear conditioning by proxy strongly depends on the social interaction between the demonstrator and the observer and necessitates an increased and tone-dependent interaction during the FCbP session. In contrast, the β2-ΚO mice are unable to modify their social behaviour and maintain low and stable interaction levels that are independent of the CS. **These data suggest that only observers that adopt a tone-dependent interaction pattern are able to acquire a socially transmitted fear response, and that this capability strongly depends on high-affinity β2-containing AChRs.**

## 3. Discussion

Learning to fear cues that predict danger enables adaptive behaviour, and acquiring information about potentially harmful stimuli from others provides a safer way to react rapidly to unfamiliar conditions. Whereas the neural circuitry of fear learning through classical conditioning has been investigated in considerable detail, the mechanisms underlying socially transmitted fear learning are less well understood. In the present study, we examine for the first time the role of high-affinity nAChRs in observational fear learning.


**Male WT mice can acquire FCbP, but only if they exhibit a flexible pattern of interaction.**


By employing the fear conditioning by proxy paradigm, we demonstrated that mice of the sociable B6 strain can acquire fear of a neutral stimulus through social interaction with a familiar FC conspecific in the presence of that stimulus. In other words, a cage mate’s conditioned response to a tone can affect the observer’s subsequent behaviour to that tone. To our knowledge, this is the first demonstration that fear conditioning by proxy is possible in mice, contradicting a previous study who suggested that prior fear experience is necessary for FCbP animals to exhibit freezing to a tone [10].

At the same time, we find that this ability does not extend to all the WT observers but only those that (a) interacted more with the FC mouse during the tone presentations and (b) manifested a tone-dependent increase in social interaction, i.e., they showed enhanced interaction with their cage mate during the tone phase compared to the pre-tone phase.

The contribution of increased overall interaction in FCbP was also highlighted in a previous study in rats [11], showing that about half of the observer rats displayed freezing to the tone, and that the amount of social interaction between FC and FCbP animals during the tone (Day 2) accounted for 28.5% of the variance in that group (Day 3). Our results are in broad agreement with this study, since we find that 12.7% of the variability is related to the duration of interaction during the tone, but further reveal that the most important predicting factor is the *pattern* and not the amount of interaction. Specifically, we show that the bulk of the variability in freezing response (69.7%) was accounted for by whether the animals were able to modify their behaviour after the first tone and increase their attempts to interact with the FC cage mates.

This novel finding—that the first tone induces an increase in social interaction—can be interpreted in two ways: either as a behaviour motivated by the conspecific’s distress: upon CS presentation, FC mice exhibit enhanced freezing, and this change in demonstrators’ behaviour is the factor that attracts the observers’ attention, or as a behaviour triggered by the tone itself: upon hearing the novel cue, observer animals become more motivated to explore their environment in order to acquire relevant information. Indeed, our experimental protocol was designed to encourage social interaction, since it included the habituation of observers to the context (chamber). The two alternatives are not mutually exclusive and suggest that this subgroup of observers (FCbP+) manage to associate the distress of their conspecific with the tone.


**Why do only a subgroup of WT observer mice acquire FCbP? Possible underlying mechanisms:**


The fact that only one third of WT observers manifested a tone-dependent interaction can be attributed to factors that regulate social fear learning. One such factor is dominance ranking, which is often established among males during the 2–4 weeks of triad housing. Indeed, a previous study in rats demonstrated that, besides social interaction, the dominance relationship between the observer and the demonstrator was crucial in determining whether fear was transmitted socially [5]. Only FCbP subordinate rats that interacted with higher-ranked FC cage mates on day 2 froze during tone presentation on day 3, and the dominant rats did not learn to fear a cue from either subordinate. While we fully intended to examine this factor in our study, the dominance status in some triads was highly ambivalent and precluded this investigation. Future work with larger sample sizes could be useful in exploring the effect of dominance status on the social transfer of fear as it could affect either the level/pattern of interaction between observers and demonstrators and/or the significance attributed to the social information [5,29]. 

Alternatively, the ability to learn through social interactions could reflect individual variations in learning strategies [30,31,32]. In our experimental protocol, the mice had to detect and integrate social signals in the absence of any direct aversive consequences. When the outcome is ambiguous, animals may have to switch between learning strategies in order to determine the most appropriate response. For instance, observers may decide to value the direct over the vicarious experience, or vice versa. This requires a behavioural flexibility that is cognitively demanding and may not be possible for all animals. Hence, the inability of FCbP- mice to acquire fear through social learning might reflect individual variations in higher cognitive functioning (e.g., strategy-shifting abilities).

Consistent with this scenario is the total inability of β2-ΚO mice to acquire FCbP (as illustrated in Figure 7B and Figure 9). Indeed, mice lacking the β2-subunit of nAChRs have both morphological alterations in the prefrontal cortex (PrL and ACC) and impaired behavioural flexibility [16,17,21,22,27,28]. These mice display impairments in tasks involving motivation ranking that reflect a difficulty in evaluating information and choosing or switching between different actions [28]. 


**Why are β2-ΚO mice unable to acquire FCbP?**


In our study, none of the β2-KO observers expressed freezing to the tone, suggesting that they failed to acquire fear through social interaction. This result cannot be attributed to the impaired fear learning of the demonstrators, since the β2-KO FC mice exhibited normal levels of fear acquisition. This finding is consistent with a previous study showing that adult β2-KO mice do not exhibit impairments in either contextual or tone-conditioned fear [33], indicating they have intact associative learning and fear memory formation/retrieval. Furthermore, β2-KO FC mice appear to transmit the same fear information during the FCbP session as WT FC mice, based on the freezing response of FC mice. Having said that, it is well known that there is a broad range of fear responses. Besides freezing, conditioned animals emit auditory and olfactory signals of fear, such as ultrasonic vocalizations (USVs) and stress-induced anxiogenic pheromones, respectively, and both of these responses to CS have been shown to affect cued fear learning in observer mice [8,34]. However, a recent study showed that β2-KO mice display a normal interest in social olfactory cues such as pheromones (consistent with our data showing no significant difference in anogenital sniffing between observers of the two genotypes), and they have no auditory deficits and do not display impaired USV emission [35]. Accordingly, there are no indications that the KO observers were exposed to different fear information during FCbP compared to the WT mice.

Our investigation of the possible factors contributing to the failure of the β2 KO mice to acquire/exhibit observational fear learning revealed that these mice displayed stable levels of social interaction with the demonstrators throughout the FCbP session. In other words, all the KO FCbP mice displayed a tone-independent interaction pattern similar to WT FCbP− animals. This finding further supports the contribution of a tone-dependent interaction pattern in FCbP. We cannot attribute this result to impaired motivation for social interaction since the β2 KO mice exhibited normal levels of interaction during the pre-tone phase (also consistent with the literature) [16,35]. On the contrary, they failed to adapt their interaction to the special conditions during the tone phase, and they did not increase their interaction, as the WT observers did. This finding indicates a possible role of β2 nAChRs on social interaction—the factor that affected observational fear learning in our study. This impairment in the β2-KO mice could be explained by a more rigid social behaviour [28], an attentional deficit [19,36], or a lack of empathy [37,38]. 


**Involvement of prefrontal cortical areas in FCbP**


Social behaviour is a defining mammalian feature that integrates emotional and motivational processes with external rewards. It is thus an appropriate readout for complex behaviours. Social interaction requires sequences of action in which the individual must adjust rapidly to the behaviour of others, organize the succession of its actions, and inhibit habitual or inappropriate responses. In other words, normal social interaction requires a range of intact executive functions, such as planning complex behaviours, inhibitory response control, decision making and behavioural flexibility. The prefrontal cortex (PFC) is recognized as a crucial brain structure for executive functions. It mediates impulsivity and compulsivity control and guides behaviour by associating sensory processing, memory and emotions [14]. The ACC is also essential for observational fear learning [6] and fear conditioning by proxy [5]. 

There is considerable evidence that β2-nAChRs play a pivotal role in cognition and executive behaviours [16,17,18,39,40] through neurotransmitter release in the PFC [for a review; 14,15]. As Dos-Santos Coura and Granon (2012) suggested, β2-nAChRs are likely to be “crucial for decision-making processes during which there is integration of emotional and motivational features for adapted and flexible goal-directed behaviors” [14]. Data indicating that β2-nAChRs are directly implicated in behavioural flexibility [16,20,21,28,35] are consistent with our findings on the failure of the β2-KO mice to enhance interaction during the special conditions they were exposed to during the tone phase of FCbP. Therefore, we suggest that the rigid social behaviour displayed by the β2-KO FC mice is the most parsimonious explanation for their failure to acquire cued fear through social observation.

## 4. Materials and Methods

### 4.1. Animals and Housing

The study was performed in the animal facility of the Centre for Experimental Surgery of the Biomedical Research Foundation of the Academy of Athens and was evaluated and authorized by the Veterinary Service of the Prefecture of Athens, as required by the Greek legal requirements for animal experimentation.

A total of 84 male C57BL/6J mice (B6) at the age of 3.5 to 4.5 months old were studied (body weight: 25–35 g). In more detail, the sample consisted of 42 β2 knockout (β2-ΚO) and 42 wild-type (WT) mice. All animals were obtained from the breeding colony of the animal facility of the Foundation and were housed at a room temperature of 24 ± 2 °C, a relative humidity of 55 ± 10% and a 12 h:12 h light/dark cycle (07:00/19:00). Animals were maintained according to the Guide for the Care and Use of Laboratory Animals and the relevant recommendations of the European Commission on the care and use of laboratory animals.

Fifteen days prior to the experiment, the animals were randomly divided into triads and housed in H-Temp™ polysulfone type III cages (365 mm (L) × 207 mm (W) × 185 mm (H), H-Temp™, Tecniplast, Buguggiate, Varese, Italy). The bedding in each cage comprised corncob bedding (Rehofix MK 2000, J. Rettenmaier & So, Rosenberg, Germany). The cages were cleaned once a week. All animals had ad libitum access to filtered tap water in drinking bottles and a pelleted chow that contained 18.5% protein, 5.5% fat, 4.5% fiber, 6% ash (irradiated vacuum packed, 2918, Harlan, Italy).

### 4.2. Experimental Design

Animals were housed in triads for 2–4 weeks prior to the beginning of the experiment. Each triad comprised (a) a demonstrator that would undergo classical fear conditioning (i.e., the fear-conditioned animal, or FC), (b) an observer that would undergo fear conditioning by proxy (i.e., the fear-conditioned-by-proxy animal, or FCbP) and (c) a naïve control. The animals of each cage were randomly marked as FC, FCbP and N the first day of the experiment. All mice were habituated to the behavioural testing room, the transport and the experimenter for 5 days prior to the beginning of the experiment. All the behavioural tests were conducted during the light phase of the animal light/dark cycle (07:00–19:00) at room temperature of 22 ± 1 °C and low-level illumination (30 lux).

To test our hypotheses, we conducted the fear conditioning by-proxy paradigm, a 3-day experimental protocol presented concisely in Figure 10.

**Figure 10 ijms-24-15143-f010:**
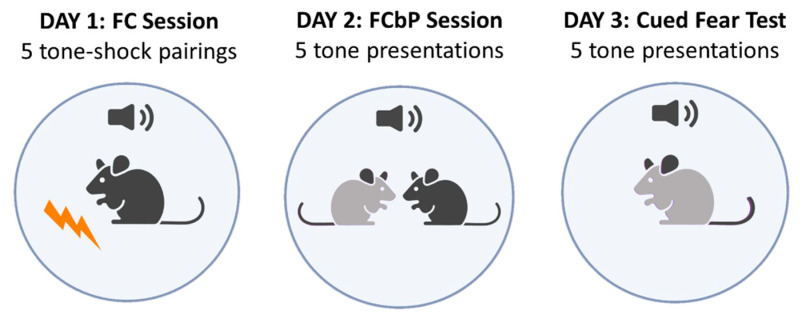
The Fear Conditioning by Proxy protocol. On Day 1, the demonstrator of the triad underwent fear conditioning to a tone (FC, black). Twenty-four hours later, the FC animal was placed back to the conditioning chamber with its observer cage mate (FCbP, grey), and they were presented with the tone. In this session, the FCbP mouse experienced the conditioned responses of the FC animal to the tone. Twenty-four hours later, the FCbP mouse was tested for fear expression to the tone.

### 4.3. Apparatus

Each session of the FCbP paradigm was performed in a two-compartment shuttle box (590 (W) × 190 (D) × 240 (H) mm; Panlab, LE 918; Harvard Apparatus Ltd., Holliston, MA, USA), which was enclosed in a second—sound attenuating—box. A fan mounted on the left wall of the sound attenuating box produced white noise and provided ventilation.

The left side of the shuttle box comprised the fear conditioning chamber (conditioning context). Fear conditioning (Day 1) and fear conditioning by proxy (Day 2) were performed in this context. This chamber was equipped with a stainless-steel grid floor connected to a shock generator (Panlab, LE 100-26), a white-light lamp connected to the shuttle box control unit (Panlab, LE 900) and a sound generator in the back wall that delivered the tone. The operation of the shuttle box (automated tone production and delivery of foot shocks) was controlled with the software program ShutAvoid v.1.8.2. (Harvard Apparatus Ltd., Holliston, MA, USA). The conditioning chamber was cleaned thoroughly with 70% ethanol before the introduction of each mouse.

To test for fear responses to the tone alone (Day 3, cued fear test), we needed to introduce the animals in a context that had not been associated with shocks. The right side of the shuttle box comprised the new context where each animal was tested for fear expression to the tone. Since this side was similar to the first one, we placed a white paperboard (a) over the grid floor, (b) diagonally in the chamber in order to alter its dimensions and (c) on the walls’ surface to differentiate them from the black walls of the conditioning context. Furthermore, we decreased the level of illumination (lux) in the shuttle box and used 70% acetic acid (instead of ethanol) to clean the chamber between trials.

The position of the mouse in the apparatus was detected by weight transducers located below the grid floor in the two chambers. Behaviour was digitally recorded throughout each session using a camera (Panasonic, CCTV, WV—BP 332EE) mounted on the top of each chamber. All parameters were measured using Ethovision XT8.5 specialized video tracking software.

### 4.4. Procedure

*Day 1: Fear Conditioning.* The demonstrator of each triad was individually placed in the conditioning chamber and underwent fear conditioning to a tone [in accordance with published methods for classical fear conditioning; [33,41]. As shown in Figure 11A, the animal was habituated to the conditioning chamber for 10 min and then received five tone–shock pairings: each tone (30 s, 1 kHz, 85 dB) co-terminated with a foot shock (2 s, 0.5 mA). The interval between two consecutive tones (inter-trial interval, ITI) was 2 min. The FC animal returned to its cage 30 s after the last pairing, and the animal’s behaviour was recorded throughout the process. Afterwards, the observer (FCbP) animal of each triad was habituated to the conditioning chamber for 10 min, and right after was placed back in the cage.

*Day 2: Fear Conditioning by Proxy*. One day after conditioning, the demonstrator and the observer of each triad were placed together in the conditioning chamber for the fear conditioning by proxy session (Figure 11B). The animals were habituated to the chamber for 10 min and then received five tone presentations (no shock, 2 min ITI). They could freely interact and their behaviour was recorded throughout the session. The animals returned to their cage 30 s after the last tone presentation. Subsequently, the naïve control of the triad was separately habituated to the conditioning chamber (10 min) and returned to the cage.

*Day 3: Fear learning tests*. Twenty-four hours later, each mouse of the triad was tested separately for freezing (fear response) to the tone (Figure 11C). The mouse was placed in a new context, and after 3 min, it received five tone presentations (30 s, 2 min ITI). The animal’s behaviour was recorded throughout the session, both during the tone presentations and during the ITIs.

**Figure 11 ijms-24-15143-f011:**
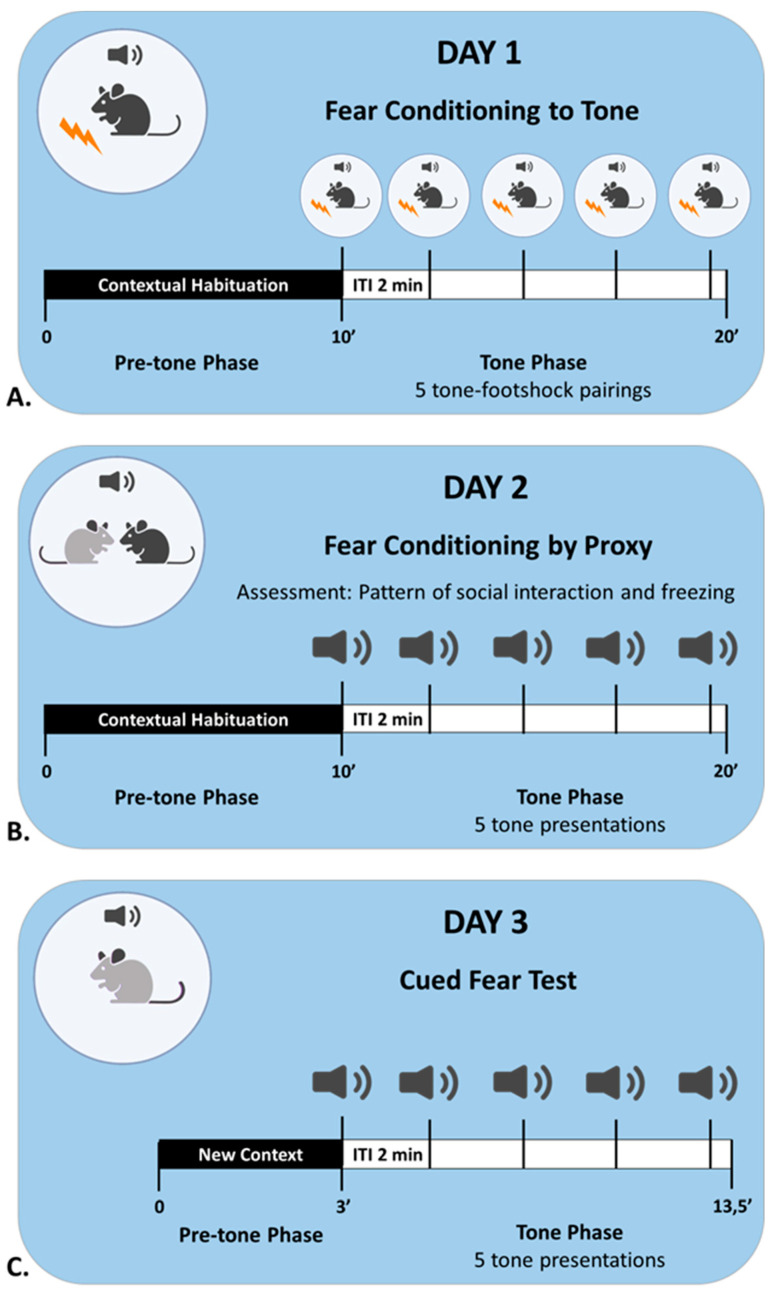
(**A**). The fear conditioning session on day 1 consisted of the pre-tone and tone phases. During the pre-tone phase, the demonstrator was habituated to the conditioning context. In the tone phase, the animal received five tone–shock pairings, with 2 min intervals between pairings. (**B**). The fear conditioning by proxy session on day 2 differed from the FC session in two ways: the observer and the demonstrator were placed together in the conditioning chamber, and there were only five tone presentations (no shock). (**C**). For the test session on day 3, each animal was examined individually for cued fear in a new context. Freezing behaviour was recorded both during the pre-tone phase (3 min) and during the tone phase (5 tones + 4 ITIs).

### 4.5. Behavioural Scoring

During each test session, we recorded the animals’ behaviour. Two individual raters blind to the experimental conditions manually scored *freezing* and *interaction*.

***Freezing.*** Freezing constituted the fear index and was defined as the absence of any movement—except breathing—for at least 1 s. It was expressed as the percentage of time the animal spend immobile during the (1) pre-tone phase (habituation), (2) entire tone phase (tone presentations and ITIs) or (3) tone presentations alone (30 s ∗ 5 times; 150 s).

***Interaction.*** Interaction was expressed as the percentage of time that the observer spent in contact with the demonstrator during (1) the whole session, (2) pre-tone phase/habituation, (3) entire tone phase (tone presentations and ITIs) or (4) tone presentations alone (30 s ∗ 5 times; 150 s). Social contact was defined as any physical contact or interaction (qualitatively defined below), excluding accidental contact made in passing. This contact comprised seven unique behaviours that the observer directed towards the demonstrator: allogrooming, paw contact, body contact, anogenital sniffing, nose-to-nose contact, play and rattling observation. *Allogrooming* occurred when the observer groomed (licked) the demonstrator. *Paw contact* occurred when the observer placed one or both of his paws on the demonstrator (excluding both accidental contact from trying to get around the demonstrator or using the demonstrator as a support to reach a different area of the chamber). *Body contact* occurred when the observer maintained close contact with the demonstrator by either leaning against the demonstrator or huddling against him. Anogenital *sniffing* occurred when the observer actively sniffed at the genital area of the demonstrator. *Nose-to-nose contact* occurred when the two mice touched noses while facing one another. *Play* occurred when the two mice engaged in any mode of playful behaviour, including wrestling, pouncing, biting, or chasing. *Rattling observation* occurred when the observer approached and actively observed (stretched toward) the demonstrator’s tail while the demonstrator displayed tail rattling. Tail rattling constitutes an eye-catching fast movement of the tip of the tail, which makes a characteristic sound and is thought to be a threat behaviour [42]. Each social contact type was expressed as (1) the number of contacts encountered and (2) the percentage of time the observer displayed the contact during each interval of interest.

### 4.6. Statistical Analysis

The statistical software IBM SPSS Statistics for Windows (SPSS Version 26.0 Armonk, NY, USA: IBM Corp) was used for all statistical analyses. The Shapiro–Wilk test was used to assess normality (Gaussian-shaped distribution) for all continuous variables, and any outliers were detected by inspecting the boxplots. When needed, the equality of error variance was assessed by Levene’s test. Since no “play” contact was encountered and “rattling observation” was vague or difficult to assess for some of the mice, these two variables were excluded from the analysis of interaction on day 2. Regarding the analysis of freezing time on the cued fear test (day 3), two WT-FCbP and two WT-Naïve animals were excluded due to health issues, leading to a sample size of 12 WT animals in each group instead of 14. In the same analysis, the dependent variable of freezing time was not normally distributed in most of the experimental groups, so the data were transformed as their squared root. The statistical analysis was carried out on the transformed data, while the graphs show the original data for ease of presentation. Results are presented as means ± standard error. For all comparisons, statistical significance was set at *p* < 0.05.

## Figures and Tables

**Figure 1 ijms-24-15143-f001:**
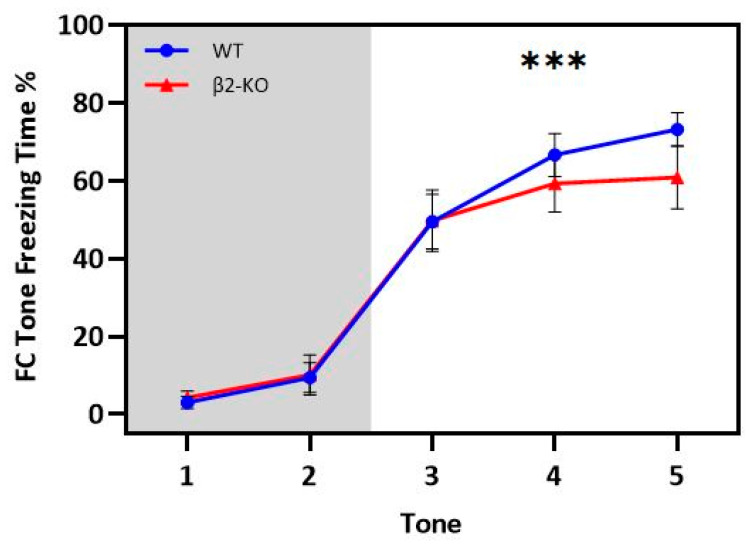
Percentage of freezing time of the FC animals during tone presentations on day 1. There are no significant differences in the freezing pattern between WT and β2-ΚO mice. There is a significant (***) increase in freezing after the presentation of the 3rd tone for both genotypes.

**Figure 2 ijms-24-15143-f002:**
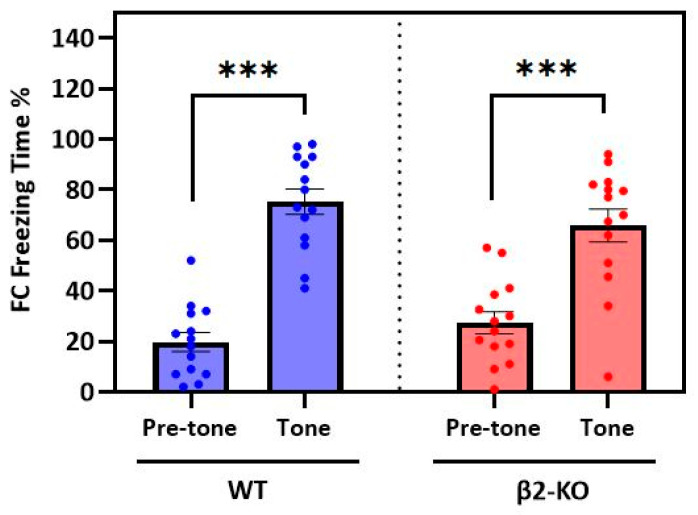
Mean percentage of freezing time of the FC animals during pre-tone and tone phases on day 2. There is no significant difference between the WT and β2-ΚO groups, and both genotypes froze significantly more (***) during tone phase compared to the pre-tone phase.

**Figure 3 ijms-24-15143-f003:**
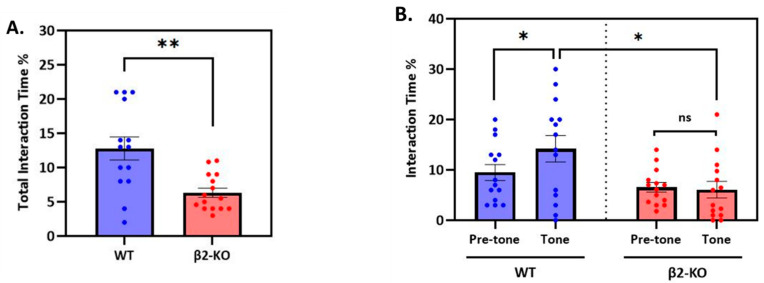
Patten of social interaction between FC and FCbP mice on day 2. (**A**) Total interaction time over the entire FCbP session: the WTs interacted significantly longer (**) than the KOs. (**B**) Phase-dependent interaction: WTs -but not KOs- exhibited a tone-dependent social interaction (*) and interacted significantly more (*) than KOs during tone phase.

**Figure 4 ijms-24-15143-f004:**
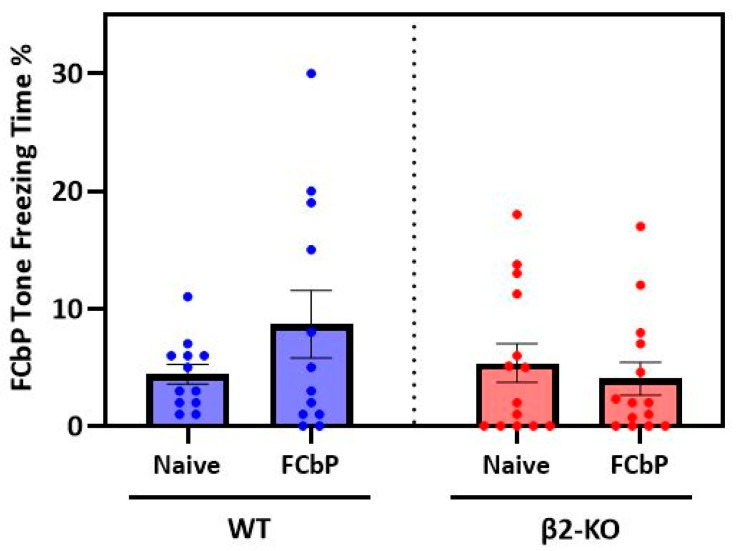
Percentage of freezing time of the FCbP and naïve animals during tone presentations on day 3. There is no significant difference in freezing behaviour either between genotypes or between groups.

**Figure 5 ijms-24-15143-f005:**
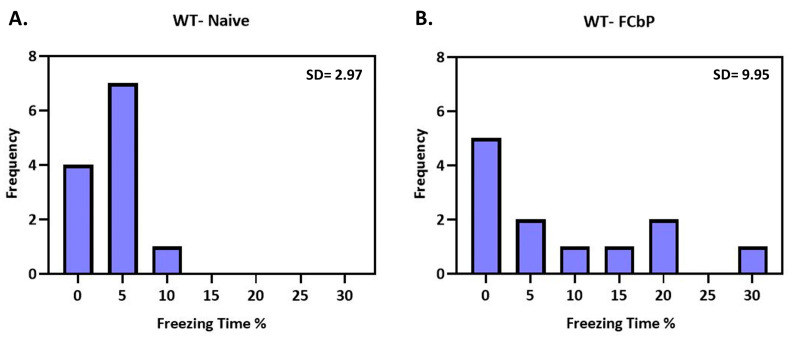
Distribution of the freezing times of (**A**) naïve and (**B**) FCbP, WT animals during day 3 showing that WT-FCbP animals have a greater variance compared to naïve controls.

**Figure 6 ijms-24-15143-f006:**
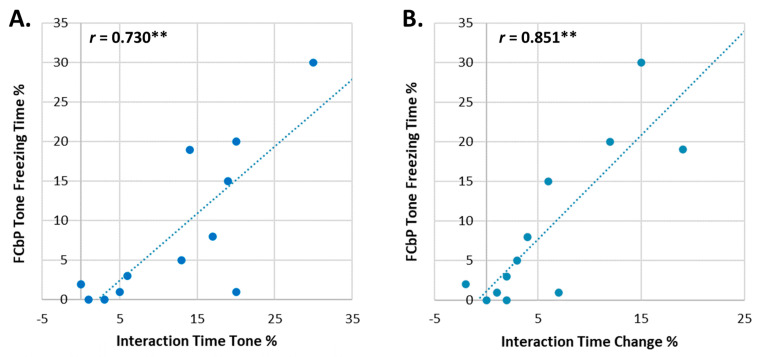
Scatter plots of the two predictor variables that significantly explain the bulk of the variation in WT FCbP freezing time during tone presentations on day 3. (**A**) The interaction time between FCbP and FC mice during tone presentations and (**B**) the change in interaction time between the pre-tone and tone phases, have a significant positive Pearson’s correlation (**) with the freezing time displayed by FCbP WT animals on day 3.

**Figure 7 ijms-24-15143-f007:**
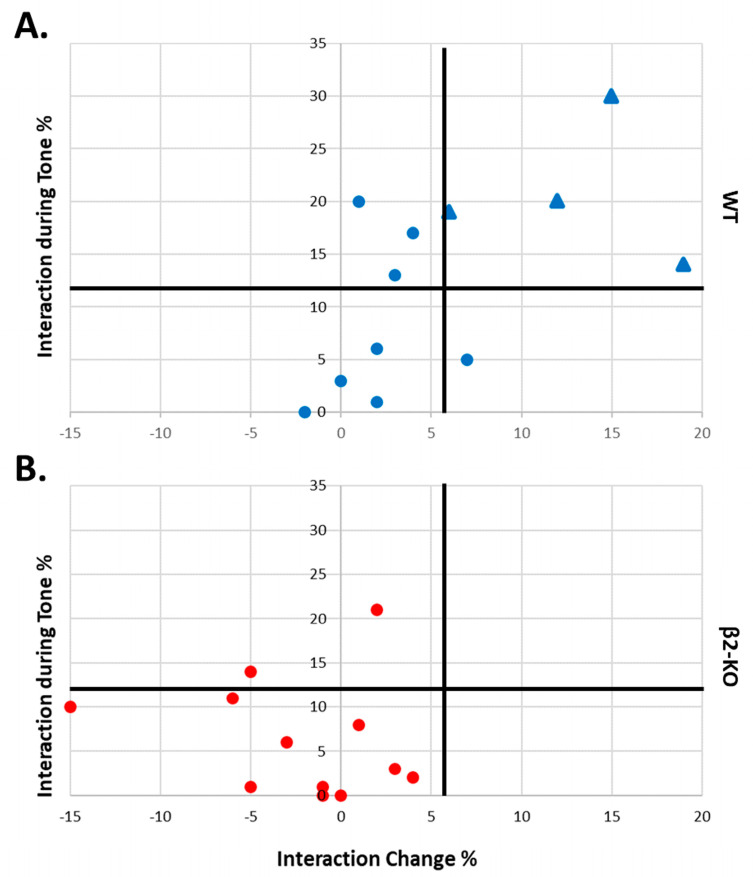
Diagrams illustrating the interaction patterns in FCbP animals on Day 2. The horizontal and vertical black lines represent the mean interaction during the tone and mean interaction change, respectively. (**A**) WT mice that displayed high levels of interaction during tone presentations and increased interaction after the first tone (interaction change) are shown in the upper right quadrant (blue triangles) and comprise the FCbP+ group. All the other WT mice that do not fulfil these two criteria are shown in the other three quadrants (blue circles) and comprise the FCbP− subgroup. (**B**) None of the β2-ΚO mice exhibit this tone-dependent interaction behaviour (red circles).

**Figure 8 ijms-24-15143-f008:**
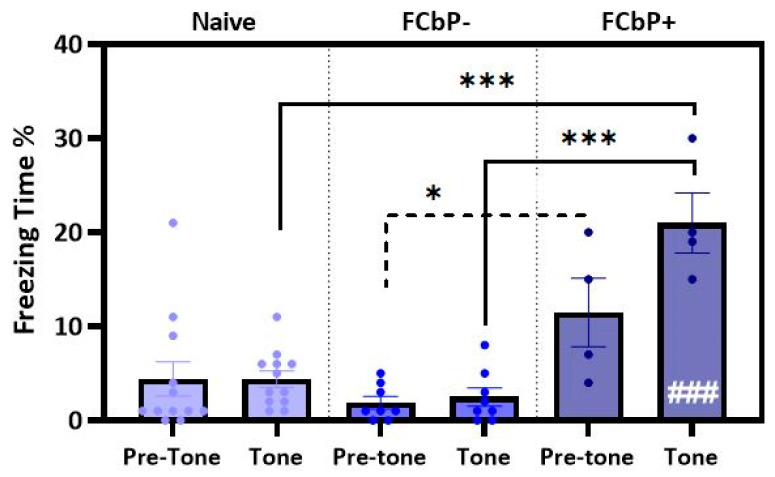
Conditioned fear response to the tone displayed by the two WT FCbP subgroups (FCbP+: tone-dependent interaction pattern, FCbP−: tone-independent interaction pattern) compared to naïve animals. FCbP− and naïve WT mice showed no differences in freezing times between the pre-tone phase and the tone presentations. Instead, WT-FCbP+ mice exhibited a significant increase in freezing between pretone and tone (###) and froze more than either naïve or FCbP− mice (***). The WT-FCbP+ group also froze more than the WT-FCbP− group during pre-tone phase (*, dotted line).

**Figure 9 ijms-24-15143-f009:**
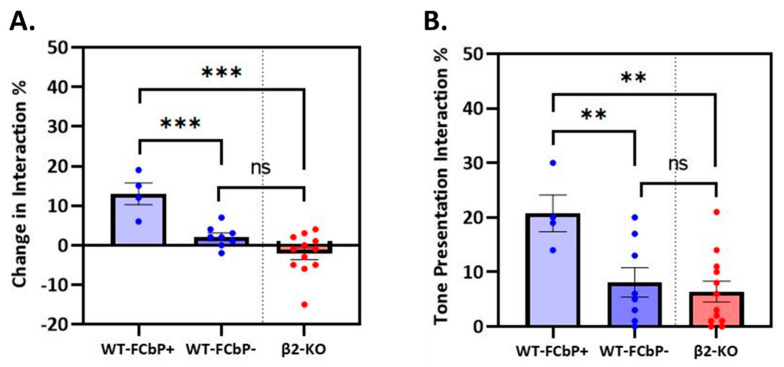
Comparison of social interaction pattern between the three FCbP groups. (**A**) Change in interaction between pre-tone and tone phase. Only the WT-FCbP+ group displayed increased interaction during tone phase (***). (**B**) Levels of interaction during tone presentations. The WT-FCbP+ group displayed significantly higher interaction during tone presentations compared to both WT-FCbP− and β2-ΚO FCbP animals (**).

**Table 1 ijms-24-15143-t001:** Potential factors contributing to the freezing variation displayed by FCbP mice.

FCbP Freezing (Criterion Variable)	Social Interaction Pattern (Predictors)
Freezing time during tone (%)	1. Interaction in FCbP session
	2. Interaction in pre-tone phase
	3. Interaction in tone phase
	4. Interaction change (3–2) *
	5. Interaction during tone *
	6. Interaction during ITIs
	7. Social contacts (total number of all contacts)
	8. Nose contact
	9. Paw contact
	10. Body contact
	11. Anogenital sniffing
	12. Allogrooming

## Data Availability

All data included in this study are stored at BRFAA archives and can be made available upon request.

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
