# Peer review of "Fear Conditioning by Proxy: The Role of High Affinity Nicotinic Acetylcholine Receptors"

_ijms, 2023, doi:10.3390/ijms242015143_

Round 1
Reviewer 1 Report
The paper describes that a subset of wild type mice appears to learn fear conditioning from a conspecific and that the likelihood that this occurs increases with more social interaction between the pair on day 2. This effect is not seen in beta2 AChR knockout mice. Overall, the paper is well written, however, it does not include much consideration of why only one third of WT mice display the behaviors.
Was there any attempt to classify mice by their hierarchical behavior – dominant vs submissive - to further probe why there was a difference in the WT group?
Was the type of social interaction similar or different? Although this was not statistically significant in the regression, were there specific behaviors or groups of behaviors that were more often apparent during the epochs of interest?
Use of stepwise regression is not universally accepted and should be justified further.
Paragraph beginning line 472 and Fig 10 are unnecessary as the FCbP+ group was selected based on their conditioned response. The information can be used to support the group separation described in the previous paragraph.
Author Response
The paper describes that a subset of wild type mice appears to learn fear conditioning from a conspecific and that the likelihood that this occurs increases with more social interaction between the pair on day 2. This effect is not seen in beta2 AChR knockout mice. Overall, the paper is well written, however, it does not include much consideration of why only one third of WT mice display the behaviors.
We thank the reviewer for this comment, which encouraged us to enrich the discussion section of our paper. The fact that only one third of the observers manifested flexible pattern of interaction can be attributed to factors that regulate social fear learning. It is highly likely that dominance ranking, which may be naturally established among males during the 2-4 weeks of triad housing, contributes to the two observed freezing patterns. Jones and Monfils (2016) demonstrated that besides social interaction, the dominance relationship between the observer (FCbP) and the demonstrator (FC) rat was crucial in determining the amount of fear that was transmitted socially, as only FCbP subordinate rats that interacted with higher ranked FC cagemates on day 2, froze during tone presentation on day 3. Dominant rats did not learn to fear a cue socially from either subordinate. Thus, it would be interesting to explore the effect of dominance status in social transfer of fear, as it could affect either the interaction between observers and demonstrators and/or the significance attributed to the social information (Kavaliers et al 2005; Jones and Monfils, 2016). Alternatively, the different pattern of social interaction could reflect individual variations in social anxiety, exploratory behaviour, or learning strategies, not directly associated with social status (Galef, 2009; Rendell et al., 2011; Mesoudi et al., 2016). For instance, in our fear conditioning by proxy paradigm mice had to detect and integrate social signals in the absence of any direct aversive consequences. When the outcome is ambiguous, animals may have to switch learning strategies in order to determine the appropriate response. In this way, fear conditioning by proxy could assist the investigation of higher cognitive functioning.
Following the reviewer’s comment, we have now included these considerations in the revised manuscript (highlighted, pages 16-17).
Was there any attempt to classify mice by their hierarchical behavior – dominant vs submissive - to further probe why there was a difference in the WT group?
We thank the reviewer for this suggestion. Indeed, we were aware of the significance of the hierarchy in mouse behaviour (Lathe, 2004; Kunkel and Wang; 2017) and we intended to examine its possible effect in social transmission of fear, by classifying mice of each triad by their hierarchical behavior. Unfortunately, the classification was highly ambiguous for some triads, and thus we could not obtain a clear effect of dominance. We strongly believe that dominance ranking should be considered in this line of research in mice, especially since it has been shown that it is a significant predictor of fear conditioning by proxy in rats (Jones and Monfils 2016).
Following the reviewer’s comment, we have added this in the discussion (highlighted, page 16).
Was the type of social interaction similar or different? Although this was not statistically significant in the regression, were there specific behaviors or groups of behaviors that were more often apparent during the epochs of interest?
Interestingly, there was no statistical difference in the type of social interaction between the two FCbP subgroups. We also did not notice any preferred social behaviour during the epochs of interest. While observers attempted several possible types of interaction, the demonstrators after the first tone presentation were often very still which precluded further interaction.
Use of stepwise regression is not universally accepted and should be justified further.
We thank the reviewer for this comment. The main reason we used a stepwise regression model is to have a direct comparison to previous publications using the FCbP protocol on which we based our study (Bruchey et al. 2010; Jones et al. 2014; Jones and Monfils 2016). In all these studies, regression models indicate that social contact is the strongest predictor of the freezing variation.
Following the reviewer’s comment, we have added this information (highlighted, page 11).
Paragraph beginning line 472 and Fig 10 are unnecessary as the FCbP+ group was selected based on their conditioned response. The information can be used to support the group separation described in the previous paragraph.
We thank the reviewer for this comment, which allows us to clarify an important misunderstanding. The separation of the FCbP group was not based on the conditioned response, but on the interaction pattern exhibited on Day 2. In more detail, observers that adopted a “tone-dependent interaction pattern” (enhanced social interaction during the tone phase AND high interaction levels during tone presentations) were the FCbP+ subgroup. On the other hand, observers that displayed a tone-independent interaction pattern, with quite stable or low interaction (low “interaction change” AND/OR low “interaction during tone”) comprised the FCbP- subgroup. Based on the regression model, we then hypothesized that the FCbP+ subgroup would freeze more to the tone than FCbP- subgroup and/or the control group, and we proceeded to test it and investigate if this difference was indeed significant. Therefore, we believe it is essential to include this paragraph, as well as Fig 10.
Following the reviewer’s comment, we have rewritten this section (highlighted, page 12-13).

Reviewer 2 Report
The manuscript is clearly written and well organized.
One minor point for the authors to consider: Will beta-2 mice be able to express fear conditioning if they had observed a demonstrator mouse receiving repeated cs-us?
Author Response
The manuscript is clearly written and well organized.
One minor point for the authors to consider: Will beta-2 mice be able to express fear conditioning if they had observed a demonstrator mouse receiving repeated cs-us?
We thank the reviewer for this interesting question. This is an issue we have not yet examined, but we can speculate a little. Using an observational fear learning model, Allsop et al. (2018) have demonstrated a causal relationship between the transfer of socially extracted information from the ACC to the BLA and the ability of mice to learn about dangerous stimuli in their environment through observation. Interestingly, inhibition of ACC input to the BLA did not impair classical fear conditioning, suggesting that this pathway is only required for observational learning, but not for associative learning in general. Also, in the fear conditioning paradigm in rats, Jones and Monfils (2016) indicated that the ACC is necessary only for the social transmission of fear, but not required for classical fear conditioning.
Mice lacking β2-containing receptors display morphological alterations in neurons of the cingulate cortex (CC; Konsolaki and Skaliora, 2015; Ballesteros-Yanez et al., 2010) that may mediate the impaired social transmission of fear. It would indeed be interesting to examine the differential effect of observing a familiar β2 knockout mouse undergoing classical fear conditioning vs. interacting with a fearful FC β2 knockout mouse on social fear learning, as this could provide insights on the role of β2-containiang nAChRs. Further investigating the underlying neural processes of the two models of social fear learning could also unravel the degree of overlap in the brain circuits that underlie these processes.
Reviewer 3 Report
In the manuscript entitled "Fear conditioning by-proxy: The role of high affinity nicotinic acetylcholine receptors", Chalkea et al. studied the effect of observational fear-learning in wild-types (WT) and mice lacking β2-subunit of the nicotinic acetylcholine receptor (β-KO). They noted that, among triads housing WT mice, when observer mice put together with conditioned mice, showed social transmission of fear depending on the interaction pattern between animals during cue presentation. This social transmission of fear was lacking in the β-KO mice, perhaps due to rigid interaction pattern compared to WT mice. Based on the results authors concluded that, observers that adopt a tone-dependent interaction pattern can acquire a socially transmitted fear response and this capability strongly depends on high affinity β2-containing AChRs.
The current study has succeeded in putting forth the influence of observational fear learning in mice. Use of the β-KO mouse-line to study the involvement of nicotinic Ach system in behavioral modulation has improved the significance of the study. The experiments are generally well-controlled, well-executed and the results are of potential interest to the field, however the following concerns need to be addressed.
Major concerns:
1) While authors are able to show that the interaction time is associated with observational fear learning, the major effect on freezing to the tone is non-significant in observers as compared to naïve mice (figure 6). While considering individual variability, authors have categorized these mice into the FCbP+ and FCbP- cohorts, showing significant differences between them (Figure 10). It seems that less than half (only 4 mice) were FCbP+, raising robustness of the model. In the backdrop of these results authors need to discuss validity of their model as compared to more robust models available (for example, Allsop et al, PMID: 29731170).
2) The criteria used to segregate FCbP+ and FCbP- cohorts is not clear. It should be explained explicitly in the method section.
3) Since this study uses only males, what’s the effect of observational learning in female mice, or if any sex differences exist, in this model is not clear.
4) Authors need to discuss the probable mechanism underlying the less freezing observed in the FCbP- cohort.
5) Further to confirm the role of social interaction in observation fear learning in WT mice, authors may consider testing them using a barrier between observer and conditioned mice, on day 2.
Minor comment
1. Introduction section is too long and can be reduced.
The several information in method section seems repeatitive, which can be improved.
Author Response
In the manuscript entitled "Fear conditioning by-proxy: The role of high affinity nicotinic acetylcholine receptors", Chalkea et al. studied the effect of observational fear-learning in wild-types (WT) and mice lacking β2-subunit of the nicotinic acetylcholine receptor (β-KO). They noted that, among triads housing WT mice, when observer mice put together with conditioned mice, showed social transmission of fear depending on the interaction pattern between animals during cue presentation. This social transmission of fear was lacking in the β-KO mice, perhaps due to rigid interaction pattern compared to WT mice. Based on the results authors concluded that, observers that adopt a tone-dependent interaction pattern can acquire a socially transmitted fear response and this capability strongly depends on high affinity β2-containing AChRs.
The current study has succeeded in putting forth the influence of observational fear learning in mice. Use of the β-KO mouse-line to study the involvement of nicotinic Ach system in behavioral modulation has improved the significance of the study. The experiments are generally well-controlled, well-executed and the results are of potential interest to the field, however the following concerns need to be addressed.
Major concerns:
1) While authors are able to show that the interaction time is associated with observational fear learning, the major effect on freezing to the tone is non-significant in observers as compared to naïve mice (figure 6). While considering individual variability, authors have categorized these mice into the FCbP+ and FCbP- cohorts, showing significant differences between them (Figure 10). It seems that less than half (only 4 mice) were FCbP+, raising robustness of the model. In the backdrop of these results authors need to discuss validity of their model as compared to more robust models available (for example, Allsop et al, PMID: 29731170).
We thank the reviewer for this comment, which gives us the chance to clarify some important differences between the Allsop study and ours that prevents a direct comparison. Allsop et al. (2018) indeed demonstrated that mice can acquire observational fear after observing through a transparent perforated divider a conspecific undergoing classical fear conditioning, but only if they have previously experienced themselves an unpaired shock. In contrast, observers in our experiment did not have any experience of the aversive stimulus (direct or indirect). They had merely been exposed to the distress of their familiar fear conditioned conspecific in the presence of the cue in a habituated context. Hence, the animals in our study had to detect and integrate social signals in order to adapt their behaviour in the absence of any direct aversive consequence.
Although Allsop’s observational fear conditioning model is more robust, the observers’ mean freezing time is quite low. In contrast, our model of fear conditioning by-proxy displays a bimodal freezing distribution with some mice display low freezing (similar to the naïve group), and others quite impressive freezing levels. A similar bimodal pattern has also been demonstrated in larger samples of FCbP rats (Bruchey et al. 2010; Jones et al. 2014; Jones and Monfils 2016; Jones et al. 2018).
Following the reviewer’s comment, we have now highlighted these differences and the choice of model in the revised manuscript (pages 2 and 11).
2) The criteria used to segregate FCbP+ and FCbP- cohorts is not clear. It should be explained explicitly in the method section.
We thank the reviewer for this point. This is an important step in our study so it should be totally clear to readers. Based on the results of our predictive model, wt observers that enhanced markedly the social interaction in the tone interval (high levels of “interaction change”) AND displayed high interaction during tone presentations (high levels of “interaction during tone”) on Day 2 would be expected to exhibit more freezing to the tone on Day 3. As shown in Figure 9, only one third of the observers displayed this tone-dependent interaction pattern, while the rest displayed a tone-independent interaction pattern, with quite stable or low interaction (low “interaction change” AND/OR low “interaction during tone”). We defined the former as FCbP+ and the latter as FCbP-.
Following the reviewer’s comment, we have modified the text on pages 12 and 13 to make this classification clearer.
3) Since this study uses only males, what’s the effect of observational learning in female mice, or if any sex differences exist, in this model is not clear.
We thank the reviewer for this point. It is of course true that sex differences might exist between male and female animals, as shown in recent studies on observational learning, social communication and emotional contagion in rodents (Choleris and Kavaliers, 1999; Mikosz et al, 2015; Rigney et al, 2021). Particularly, fear conditioning by proxy paradigm in rats has indicated that social contact is the strongest predictor of FCbP freezing and that there is a sex difference for when this critical contact occurs (Jones et al., 2018). The strongest predictor of social fear learning in males was the social contact during CS presentation, while in females the social contact immediately after CS presentation.
However, our model explores social transmission of fear in male B6 mice, as stated in the methods section of our paper. Following the reviewer’s comment, we have added this clarification also in the abstract and introduction (highlighted in yellow). Current experiments in our lab investigate the effect of sex and estrus cycle phase in observational learning.
4) Authors need to discuss the probable mechanism underlying the less freezing observed in the FCbP- cohort.
We thank the reviewer for this suggestion which encouraged us to enrich the discussion, by adding the following 3 paragraphs (highlighted, pages 16-17).
Why only a subgroup of WT observer mice acquire FCbP? Possible underlying mechanisms
The fact that only one third of WT observers manifested a tone-dependent interaction can be attributed to factors that regulate social fear learning. One such factor is dominance ranking, which is often established among males during the 2-4 weeks of triad housing. Indeed, a previous study in rats demonstrated that besides social interaction, the dominance relationship between the observer and the demonstrator was crucial in determining whether fear was transmitted socially (Jones and Monfils 2016). Only FCbP subordinate rats that interacted with higher ranked FC cagemates on day 2, froze during tone presentation on day 3, and dominant rats did not learn to fear a cue from either subordinate. While we fully intended to examine this factor in our study, the dominance status in some triads was highly ambivalent and precluded this investigation. Future work with larger sample sizes could be useful in exploring the effect of dominance status in social transfer of fear, as it could affect either the level/pattern of interaction between observers and demonstrators and/or the significance attributed to the social information (Kavaliers et al., 2005; Jones and Monfils, 2016).
Alternatively, the ability to learn through social interactions could reflect individual variations in learning strategies (Galef, 2009; Rendell et al., 2011; Mesoudi et al., 2016). In our experimental protocol mice had to detect and integrate social signals in the absence of any direct aversive consequences. When the outcome is ambiguous, animals may have to switch between learning strategies in order to determine the most appropriate response. For instance, observers may decide to value more the direct over the vicarious experience, or vice versa. This requires a behavioural flexibility that is cognitively demanding and may not be possible for all animals. Hence, the inability of FCbP- mice to acquire fear through social learning might reflect individual variations in higher cognitive functioning (e.g. strategy-shifting abilities).
Consistent with this scenario, is the total inability of β2-ΚΟ mice to acquire FCbP (as illustrated in figs 9B and 11). Indeed, mice lacking the β2-subunit of nAChRs have both morphological alterations in prefrontal cortex (PrL and ACC), and impaired behavioural flexibility (Granon et al., 2003; Ragozzino and Rozman, 2007; Serreau et al., 2011; Avale et al., 2011; Konsolaki and Skaliora 2015, Konsolaki et al 2016). These mice display impairments in tasks involving motivation ranking that reflect a difficulty to evaluate information and choose or switch between different actions (Serreau et al., 2011).
5) Further to confirm the role of social interaction in observation fear learning in WT mice, authors may consider testing them using a barrier between observer and conditioned mice, on day 2.
We thank the reviewer for this suggestion. Indeed, there are several possible extensions to our study and the addition of a barrier between observer and FC mice is a plausible idea. We fully intend to explore further the factors that underlie individual variability (e.g. hierarchy, empathy, etc) and also to include additional indexes of emotional response (e.g. risk assessment behaviors, autonomic response) in order to complement our current findings. This hopefully will enable us to create a more robust model of fear conditioning by-proxy.
Minor comment
- Introduction section is too long and can be reduced.
We have attempted to exclude less important parts, the introduction has now been modified and is reduced by over 30 lines.
Comments on the Quality of English Language
The several information in method section seems repeatitive, which can be improved.
We have significantly rewritten, edited and improved this section.

Reviewer 4 Report
Overall, the study is comprehensive, well-written and organized. I have no major concerns. Minor issues are as listed below.
Minor concerns:
Regarding the bimodal freezing behaviour of WT mice (line 412 – 421), I strongly suggest to include the following point of discussion: It is highly likely that dominance ranking, which is naturally established among males during the 2-4 weeks of triad housing, contributes to the two observed freezing patterns. Thus, compared to a dominant observer, a subordinate observer may spend less time interacting with a dominant FCbP demonstrator male. This in turn leads to less freezing on day 3 in the presence of the auditory stimulus. These considerations also apply to the more rigidly behaving b2-ko mice. Since this mouse line is impaired in social interaction and observational fear learning, and shows low levels interaction per se (see introduction and discussion), it may have a subordinate character to begin with, resulting in reduced interaction on day 2 and reduced freezing on day 3.
Also, does the b2-ko mouse line has C57BL/6J background? A different background may result in different behaviour.
Table 2 is confusing:
In the online version I had access to (see above), the right column is labeled “Social Interaction Pattern (Predictors)” with the 12 predictors listed, which is OK. However, the column on the left, labeled “FCbP Freezing (Criterion Variable)” and “freezing time during tone (%)”, does not have any values?
In addition, there is a discrepancy between the predictors listed in Table 2 and those explained in the method section. Only five of the seven predictors explained (lines 283 – 305) are listed in the table, i.e., “play” and “rattling” are absent.
Furthermore, why is “7. Social contact” listed as a separate item in the table? The authors explain (lines 286 – 288) “Social contact was defined as any physical contact or interaction (qualitatively defined below), excluding accidental contact made in passing. This contact comprised of seven unique behaviours …” Thus, I understood the Social contact as bin for the various behaviours.
remarks:
- Wording: I suggest refining “sniffing” with “anogenital sniffing” throughout
- Line 196: remove Greek letters και
- Line 403: closing bracket is absent
- Text legend Figure 7: turn around order of “FCbP” and “naïve” according to A and B
- Throughout the manuscript check all equal signs (=) for space before and after (see lines 421 versus 415)
- Line 561: wordy, delete “managed to”
- Line 638: exchange “face” with “case”
- Please also include the reference to Jones et al, 2018 (https://doi.org/10.1002/cpns.43). In Figure 6, the authors show the observed bimodality in rats very convincingly due to the large number of animals analyzed, which underlines the effect seen here in the mouse (even if the number is smaller).
Well-written and organized. Occasionally long sentences could be broken into two shorter ones for better understanding.
Author Response
Overall, the study is comprehensive, well-written and organized. I have no major concerns. Minor issues are as listed below.
Minor concerns:
Regarding the bimodal freezing behaviour of WT mice (line 412 – 421), I strongly suggest to include the following point of discussion: It is highly likely that dominance ranking, which is naturally established among males during the 2-4 weeks of triad housing, contributes to the two observed freezing patterns. Thus, compared to a dominant observer, a subordinate observer may spend less time interacting with a dominant FCbP demonstrator male. This in turn leads to less freezing on day 3 in the presence of the auditory stimulus. These considerations also apply to the more rigidly behaving b2-ko mice. Since this mouse line is impaired in social interaction and observational fear learning, and shows low levels interaction per se (see introduction and discussion), it may have a subordinate character to begin with, resulting in reduced interaction on day 2 and reduced freezing on day 3.
We thank the reviewer for this apt comment. Dominance ranking has been shown to affect the social transmission of fear in rats. Based on Jones and Monfils (2016), only FCbP subordinate rats that interacted with higher ranked FC cagemates on day 2, froze during tone presentation on day 3. Dominance ranking seems quite possible to have an effect in social transmission of fear in mice, too. Indeed, we fully intended to include this as a factor in our analysis, but the classification method revealed ambiguous results for some triads, and thus we could not further investigate the effect of dominance. Following the reviewer’s comment, we have included this point in the discussion (highlighted, page 16).
Also, does the b2-ko mouse line has C57BL/6J background? A different background may result in different behaviour.
We thank the reviewer for this comment. Yes, both WT and β2 knockout mice were male C57BL/6J mice (B6) at the age of 3.5 to 4.5 months old, as was indicated in the methods section (previously lines 177-179, now lines 154-156)
Table 2 is confusing: In the online version I had access to (see above), the right column is labeled “Social Interaction Pattern (Predictors)” with the 12 predictors listed, which is OK. However, the column on the left, labeled “FCbP Freezing (Criterion Variable)” and “freezing time during tone (%)”, does not have any values?
We thank the reviewer for this comment. In Table 2 the left column refers to observers freezing during tone on Day 3 (criterion variable), so it has only one entry, which refers to the variable we want to predict with the statistical model.
In addition, there is a discrepancy between the predictors listed in Table 2 and those explained in the method section. Only five of the seven predictors explained (lines 283 – 305) are listed in the table, i.e., “play” and “rattling” are absent.
We thank the reviewer for spotting the discrepancy. In the original experimental design, there were seven types of social contacts. However, none “play” contact was encountered and “rattling observation” was vague and difficult to assess for some of the mice. As a result, the actual number of social behaviors recorded was five. We have now clarified this in the revised manuscript (page 7 , line 298-300).
Furthermore, why is “7. Social contact” listed as a separate item in the table? The authors explain (lines 286 – 288) “Social contact was defined as any physical contact or interaction (qualitatively defined below), excluding accidental contact made in passing. This contact comprised of seven unique behaviours …” Thus, I understood the Social contact as bin for the various behaviours.
Each of the five social contact types was expressed as (1) the number of contacts encountered and (2) the percentage of time the observer displayed the contact, during each interval of interest. The total number of every social contact encountered was labeled as “social contact”. It was included as a potential predictor variable, since attempts to interact rather than duration of interaction could affect the observers’ freezing response.
remarks:
- Wording: I suggest refining “sniffing” with “anogenital sniffing” throughout
- Line 196: remove Greek letters και
- Line 403: closing bracket is absent
- Text legend Figure 7: turn around order of “FCbP” and “naïve” according to A and B
- Throughout the manuscript check all equal signs (=) for space before and after (see lines 421 versus 415)
- Line 561: wordy, delete “managed to”
- Line 638: exchange “face” with “case”
We are grateful to the reviewer for the careful reading. All suggested corrections have been applied to the revised manuscript.
- Please also include the reference to Jones et al, 2018 (https://doi.org/10.1002/cpns.43). In Figure 6, the authors show the observed bimodality in rats very convincingly due to the large number of animals analyzed, which underlines the effect seen here in the mouse (even if the number is smaller).
We thank the reviewer for this suggestion, which has been incorporated in the revised manuscript (page 2, line 96 & page 3 line 112 & page 11 line 429)
Well-written and organized. Occasionally long sentences could be broken into two shorter ones for better understanding.
We thank the reviewer for this recommendation, which has been incorporated in the extensively revised manuscript.
